# The Role of a Clinical Pharmacist in the Identification of Potentially Inadequate Drugs Prescribed to the Geriatric Population in Low-Resource Settings Using the Beers Criteria: A Pilot Study

**DOI:** 10.3390/pharmacy12030084

**Published:** 2024-05-28

**Authors:** Tijana Kovačević, Maja Savić Davidović, Vedrana Barišić, Emir Fazlić, Siniša Miljković, Vlado Djajić, Branislava Miljković, Peđa Kovačević

**Affiliations:** 1University Clinical Centre of Republic of Srpska, 78000 Banja Luka, Republic of Srpska, Bosnia and Herzegovina; vedrana.barisic@kc-bl.com (V.B.); sinisa.miljkovic@kc-bl.com (S.M.); vlado.djajic@kc-bl.com (V.D.); peko051@yahoo.com (P.K.); 2Faculty of Medicine, University of Banja Luka, 78000 Banja Luka, Republic of Srpska, Bosnia and Herzegovina; 3Hemofarm, 78000 Banja Luka, Republic of Srpska, Bosnia and Herzegovina; maja.savicdavidovic@hemofarm.com; 4Faculty of Pharmacy, University of Sarajevo, 71000 Sarajevo, Republic of Srpska, Bosnia and Herzegovina; emir.fazlic@medis.ba; 5Faculty of Pharmacy, University of Belgrade, 11000 Belgrade, Serbia; branislava.miljkovic@pharmacy.bg.ac.rs

**Keywords:** potentially inappropriate medication, Beers criteria, clinical pharmacist, elderly

## Abstract

Population aging is a global phenomenon. Each country in the world faces an increased number of older persons in the total population. With aging, a high prevalence of multiple chronic diseases occurs, leading to the use of complex therapeutic regimens and often to polypharmacy. Potentially inappropriate medication (PIM) is a medicine prescribed to a patient for whom the risks outweigh the benefits. Today, several tools are used to evaluate the use of pharmacotherapy in older adults, one of them is the 2019 AGS Beers Criteria. In this prospective, pilot study, we aimed to investigate if the number of PIMs in elderly patients would be significantly reduced if a clinical pharmacist performed a pharmacotherapy review. The study included 66 patients over 65 years of age who were hospitalized at the 1200-bed university hospital. The intervention was conducted by a clinical pharmacist who reviewed the patients’ pharmacotherapy and provided written suggestions to physicians. The pharmacotherapy was again reviewed at the patients’ discharge from the hospital. A total number of 204 PIMs were identified in the pharmacotherapy of the study population. At discharge, the number of PIMs decreased to 67. A total of 67% of the pharmacist’s suggestions were accepted by the physicians. The pharmacist’s intervention led to significant decrease in the number of PIMs on patients’ discharge letters.

## 1. Introduction

Population aging is a global phenomenon. In every country in the world, there is an increased number of elderly people within the total population [1]. According to data from the World Health Organization (WHO), 727 million people in the global population in year 2020 were over 65 years old. The assumption is that this number will double to 1.5 billion by 2050. Globally, the share of the population older than 65 years has increased from 6% in 1990 to 9% in 2020. This proportion is predicted to increase further to 16% by 2050, when one in six people worldwide is expected to be aged 65 or over. Globally, life expectancy has reached 72 years, where women live for an average of 74 years, i.e., five years longer than men who live for 69 years [2].

Polypharmacy is usually defined as the simultaneous use of five or more drugs and is a frequent occurrence in the elderly population [3]. Approximately 30% of people over the age of 65 use five or more medications in developed countries (High Income Countries—HICs). In HICs, people aged 65 to 74 years reported three times more adverse drug reactions, per million inhabitants per year, compared to younger people aged 5 to 19 years [4]. The manifestation of a serious adverse effect of a drug, which results in admission to hospital and death, is more common in the elderly compared to the younger population [5]. This problem seems to be more profound in low-resources settings [6]. The term hyperpolypharmacy is described as the use of 10 or more drugs and is not a rare phenomenon because the geriatric population has a high prevalence of comorbidities, and consequently they use a greater number of drugs [4].

Polypharmacy is an example of the inappropriate use of drugs. The inappropriate use of drugs is a huge problem worldwide. The WHO estimates that more than half of the prescribed and dispensed drugs are inadequate or that nearly half of them are not used properly by patients. Today, polypharmacy has become a serious problem due to its many unwanted consequences and it is increasingly present in the therapy of patients around the world. Factors that have contributed to the increasing prevalence of polypharmacy are the increase in the number of elderly people and the high prevalence of chronic diseases in old age. Important contributing factors are certainly the availability of medicines to patients and the large prevalence of self-medication. Doctors often do not have insight into all the drugs that patients use [4]. The frequency of potential drug interactions increases as the number of drugs involved in therapy increases. The risk of an adverse event was estimated at 13% for the simultaneous use of two drugs, 58% for five drugs and 82% for the use of seven or more drugs [7].

In the last few decades, numerous studies have shown a high frequency of morbidity and mortality caused by drugs. Adverse events are estimated to be the fourth to sixth leading cause of mortality in the United States, and the number of patients suffering from adverse events is even greater. In some countries, the number of hospitalizations caused by adverse drug reactions is around 10%. Treatment of the consequences of unwanted effects of drugs represents a large financial burden on the health budget. In some countries, 15–20% of the hospital budget is spent on the treatment of conditions caused by the side effects of drugs [5].

Inadequate prescribing of drugs in the elderly refers to the use of drugs when there is no clear clinical indication for them (overprescribing), the use of the indicated medicine when the risk is greater than the benefit, when a safer and more efficient alternative is available (misprescribing), potential failures in prescribing or when a drug for which there is a clear clinical indication is not prescribed (underprescribing) [8].

Potentially inappropriate medication (PIM) is medicine prescribed to a patient for whom the risks outweigh the benefits. These are also medicines whose use in elderly patients should potentially be avoided due to the high risk of unwanted reactions and/or due to insufficient evidence of their benefit [9]. Since the early 1990s, the prevalence of PIMs’ use has been examined in more than 500 studies [10]. Potentially inadequate prescribing of drugs in geriatric patients is a serious and ongoing challenge for the health system in both the most developed and developing countries. Health care for geriatric patients is complex due to physiological, pathological, pharmacokinetic and psychological changes that occur with aging [11]. In the case of the elderly, this topic is increasingly the subject of numerous studies due to the association of potentially inadequate medication prescription with the occurrence of negative clinical outcomes, i.e., the occurrence of adverse drug reactions, increased risk of hospitalization, re-hospitalization, reduced quality of life and even death [12]. Today, there are tools used to assess PIMs in elderly patients. One of the most common criteria for evaluating potentially inadequate drugs is the 2019 AGS Beers criteria [9].

According to the 2019 AGS Beers criteria, drugs are classified into five categories: potentially inappropriate medication use in older adults, potentially inappropriate medication use in older adults due to drug–disease or drug–syndrome interactions that may exacerbate the disease or syndrome, drugs to be used with caution in older adults, potentially clinically important drug–drug interactions that should be avoided in older adults and medications that should be avoided or have their dosage reduced with varying levels of kidney function in older adults. Each category provides information about the drug class, the rationale behind why the drug is potentially inappropriate, the recommendation related to each drug, the quality of evidence related to the decision to include the drug in the criteria, and the strength of the recommendation [13]. The drugs listed in these five categories are not absolutely contraindicated in elderly patients, rather they should be a subject of therapy evaluation [14].

The goal of applying the 2019 AGS Beers criteria is to improve the choice of drugs, educate clinicians and patients, and reduce the unwanted effects of drugs. They also serve as a tool for assessing the quality and costs of health care for elderly patients. For optimal use of the Beers criteria, the American Geriatrics Society recommends the application of seven key principles [15] (Table 1).

In geriatric patients, the 2019 AGS Beers criteria are widely used to identify potentially inadequate drugs and are considered one of the most common sources used to check the safety of prescribing drugs in people over 65 years of age [16]. However, the 2019 AGS Beers criteria are not a substitute for clinical assessment and individualized health care, but instead represent a guide for doctors who prescribe drugs in order to make better clinical decisions and work on the development and improvement of quality and safe use of drugs, as well as on continuous research and improvement [17].

It is well known that medication errors are a major problem in transitions of care. Since older patients account for a high percentage of these transitions, this patient group is most vulnerable to suffering from poor-quality transitional care [18]. Care transitions carry a potential risk of medication errors because of the numerous therapy changes, complicated therapy regimes and inadequate information transmission between healthcare givers [19,20]. The risk of medication errors rises with the number of medicines, hospitalization duration, age of years, female gender, more comorbidities, number of high-risk drugs, past drug allergies, poor adherence and the presence of renal or hepatic disfunction [21].

Studies from inpatient settings in HICs suggest that the inclusion of a pharmacist in the health care team could reduce mortality and improve outcomes [22,23]. Due to war devastations in our country at the end of the 20th century, clinical pharmacy started to develop with a significant time delay in comparison to developed countries worldwide (HICs). Only a decade ago did physicians gradually and occasionally begin to accept new members of the health care team when the clinical pharmacist was finally seen as a medication expert and valuable advisor on proper medications’ use. At the moment, clinical pharmacists are present on neurology, cardiology and pulmonology wards once per week, which will hopefully change as the number of clinical pharmacy specialists at the University Clinical Center of the Republic of Srpska (UCCRS) increases.

We aimed to investigate if pharmaceutical interventions on three wards of a general teaching hospital would decrease the number of potentially inappropriate medications at discharge from the hospital.

## 2. Materials and Methods

### 2.1. Study Design and Settings

This prospective, interventional, pilot study includes an analysis of pharmacotherapy in 66 patients who were hospitalized at three wards of the UCCRS: neurology, cardiology and pulmonology. UCCRS is a 1200-bed teaching hospital and serves as a regional medical center for the northern part of Bosnia and Herzegovina, the Republic of Srpska. Bosnia and Herzegovina is a post-war country which is still transitioning towards modern organized society. The health care system in Bosnia and Herzegovina is quite complex and it is divided into thirteen smaller clusters. The health care system in the Republic of Srpska is separate from the rest of the country. UCCRS is the only institution in Bosnia and Herzegovina where clinical pharmacy is implemented. The study was conducted over a period of three months (1 April to 1 July 2022) 1 and included all patients over 65 years of age who had at least one medication in their therapy. Data were recorded using a medication reconciliation form by a clinical pharmacist who made written suggestions to the physician regarding drugs that might not be suitable for the patient, using 2019 AGS Beers criteria. Moreover, the clinical pharmacist discussed possible alternatives and the best possible pharmacotherapy options for each patient with the physician. Medicines prescribed on patients’ discharge letters were recorded and compared to each patient’s pharmacotherapy before clinical pharmacist intervention.

### 2.2. Ethics

This study was approved by the institutional ethical board (No. 01-19-45-2/21).

### 2.3. Patients and Methods

The study included 66 patients hospitalized at the neurology, cardiology and pulmonology wards whose pharmacotherapy was reviewed by a clinical pharmacist during their hospital stay. Inclusion criteria were patients over 65 years of age and at least one prescribed drug in their therapy. Exclusion criteria were patients who were less than 65 years old and patients who did not have drugs in their therapy. The following data were recorded from medical history: gender, age, presenting complaint, comorbidities, medication history, newly prescribed medications during a hospital stay, medications prescribed on discharge letter, known allergies, side effects of medicines, serum creatinine level, serum albumin level, serum urea level, serum bilirubin level, serum aspartate aminotransferase level and serum alanine aminotransferase level. The assessment of the adequacy of the therapy was carried out using the 2019 AGS Beers criteria, according to which potentially inadequate drugs were identified in the examined population. The prescribed therapy for each patient was evaluated through all five categories. The intervention was carried out as medication reconciliation and submitted as a written suggestion to the doctor. The adoption of the suggestion was observed at discharge, where the number of potentially inadequate drugs was compared before and after the intervention; that is, before and after the involvement of the clinical pharmacist in the evaluation of the therapy.

### 2.4. Statistical Analysis

Descriptive and comparative statistical methods were used for data processing. The results are expressed as means with standard deviations for continuous variables and as numbers (percentages) for categorical variables. The normality of data distribution was tested using Kolmogorov–Smirnov and Shapiro–Wilk tests. Comparisons between continuous variables were performed using the Student’s t test or the Mann–Whitney test, and categorical variables were compared using the Pearson’s χ^2^ test or the Fisher’s exact test as appropriate. For a comparison of the number of PIMs before and after the intervention, the Wilcoxon test for dependent samples was used. We considered the *p* value < 0.05 to be statistically significant. The SPSS 26 statistical program was used for statistical analysis [24].

## 3. Results

The study included 66 patients, who were hospitalized at the neurology ward (N = 31), cardiology ward (N = 16) or pulmonology ward (N = 19). In order to analyze the influence of age, we divided the patients into two groups according to their age: the I age group from 65 to 75 years and the II age group over 75 years. The patients’ demographic data are presented in Table 2.

The total number of prescribed medications in the studied population was 727. The distribution of the prescribed medicines according to patients’ groups is shown in Table 3. The median number of prescribed medicines was 10 (IQR = 6) in female patients and 10 medicines (IQR = 5.75) in males. This difference was not statistically significant (*p* = 0.801). Also, the median number of prescribed medicines according to the age groups (10 drugs in both age groups) did not show statistical significance (*p* = 0.604).

The number of drugs prescribed to patients ranged from four to twenty-four per patient, as shown in Figure 1. Only one patient was prescribed four drugs by the physician, while in the other 65 patients (98.50%) polypharmacy was identified; that is, the patients were prescribed five or more drugs in their therapy. The pulmonology ward had one patient with the highest number of prescribed drugs in their therapy, a total of 24 drugs.

The total number of PIMs identified using the 2019 AGS Beers criteria after admission to the clinic was 204 (28.10%). One or more PIMs were identified in all 66 patients. The range of PIMs was from one to seven with the mean value of 3.09. In 23 patients three PIMs were identified. The largest number of PIMs, seven, was identified in two patients, while one PIM was identified in six patients. Figure 2 shows distribution of PIMs in the studied patient population.

The distribution of PIMs by wards according to the gender and age of the studied patients are shown in Table 4.

Table 5 presents a highly significant positive correlation between the number of prescribed drugs and PIMs.

The greatest prevalence of PIMs was related to medicines that affect the cardiovascular system, with a total number of identified PIMs of 91 (44.61%). Among these drugs, the loop diuretic furosemide had the highest prevalence. Table 6 shows the distribution of PIMs according to the first level of the WHO anatomical–therapeutic–chemical (ATC) system of drugs’ classification. A complete list of PIMs identified is presented in Table 7.

After the intervention of the clinical pharmacist, the total number of PIMs decreased to 67; that is, 67.10% of their suggestions were accepted by physicians. The physicians rejected the pharmacist’s suggestion if they concluded that the benefit of the medicine outweighed the risk for the patient. The difference in the number of PIMs before and after the intervention in both age groups was analyzed. The median number of PIMs before the intervention was significantly higher (three) than the median number of PIMs after intervention (two), as shown in Table 8.

## 4. Discussion

During this study, the pharmacotherapy of 66 patients was reviewed by a clinical pharmacist and 204 of the total 727 drugs (28.1%) were identified as PIMs using the Beers criteria. The number of PIMs after the pharmacist’s suggestions significantly decreased to 67 PIMs or from a median (IQR) of three (2–4) before the intervention to a median (IQR) of two (1–4) after intervention.

Perekh et al., in a multicenter, prospective, cohort study conducted in a population of 1280 patients in Great Britain, identified that 21.6% the medications were PIMs using the Beers criteria from 2015 [25]. Using the Beers criteria from 2019, which we also used, Zhang Y. et al., during research conducted in the period from 2016 to 2018, established a prevalence of PIMs of 30.05% in the examined population with 44,458 prescribed drugs [26]. In contrast to the aforementioned studies, data were also published which showed a significantly higher prevalence of PIMs compared to ours. Thus, Dan He and colleagues identified that as much as 64.8% of medications were PIMs in the elderly Chinese population, according to the Beers criteria from 2019 [27]. Also, the data from a study conducted in Serbia showed that, in a sample of 1500 cardiovascular patients, the prevalence of prescribed PIMs was extremely high and amounted to between 70.3% and 71.3% according to the Beers criteria from 2015 [28]. In our study at least one PIM was identified in all 66 patients, which clearly indicates that the prevalence of PIMs’ prescriptions in an individual patient is extremely high. The prevalence of PIMs’ being prescribed in the elderly varies significantly around the world. Díez et al., during the aforementioned research conducted in Spain, according to the Beers criteria from 2019, identified at least one PIM in 90.8% of subjects [29].

During our research, a greater number of PIMs were identified in the male population with a total of 116 drugs, while the number in the female population was 88. This difference, which was not statistically significant, can be explained by the fact that the majority of patients in the studied population were male. The results of other studies highlight the female gender as a significant risk category for PIMs, probably due to their longer life expectancy compared to men in all world populations, which is accompanied by a greater number of drugs in their therapy for various medical conditions that are a consequence of physiological and pathological processes that occur with aging [26,30,31]. The expressed rate of PIMs and the fact that at least one PIM was identified in each patient is influenced by the high prevalence of polypharmacy in the examined population (98.5%), while gender and increasing age are not of great significance. A study conducted by Khaims et al. revealed that the frequency of PIMs is not related to age and gender, but to the high prevalence of polypharmacy. In this study, positive results in the form of a reduction in the number of PIMs after the intervention of a clinical pharmacist applying the Beers criteria from 2015 in the elderly population were also obtained [9].

The medicines that affect the cardiovascular system (mainly furosemide) and medicines that affect the alimentary tract and metabolism (mainly pantoprazole) were the medication classes with the highest PIMs’ rate. According to Beers criteria from 2019, diuretics belong to a group of drugs that should be used with caution in elderly patients, because they can worsen or cause the syndrome of inappropriate antidiuretic hormone (SIADH) or hyponatremia, so sodium level monitoring is necessary prior to starting treatment or prior to every dosage change [13]. Alturki et al., in a study that included 270 patients over 65 years of age in Saudi Arabia, showed that the highest number of PIMs were proton pump inhibitors in 39.4% of study patients, which represents a much higher rate than that in our study [32]. In other studies, proton pump inhibitors were the most frequently prescribed PIMs [9,33]. According to the Beers criteria, the use of proton pump inhibitors should be avoided for a period longer than 8 weeks, except for high-risk patients such as those using oral corticosteroids or non-steroidal anti-inflammatory drugs. The reason for this use is that, with longer use, the risk of Clostridium difficile infection and bone loss or fractures increases [13].

According to the ATC classification, the group of drugs that is the third most frequently identified as being potentially inadequate is N—drugs that act on the nervous system, where the drug bromazepam (9.76%), followed by diazepam (2.44%), is the most frequently identified PIM. Benzodiazepines are very often prescribed in elderly patients, although they are known to lead to an increased risk of cognitive impairment, falls and fractures [34]. Perekh et al., already mentioned above, during the aforementioned study, had the largest number of benzodiazepines and related hypnotics (94.30%) and antidepressants (85.27%) identified [25]. The study conducted by Stojanovic M. et al. in Belgrade included 441 patients and identified the highest number of benzodiazepines (13.23%) in those over 65 years old [31].

The next group according to the ATC classification is M—drugs that act on the musculoskeletal system with a share of 13.66%, where ketoprofen (23 patients) is the third most common PIM identified in the entire study, after furosemide and pantoprazole. According to the Beers criteria, it is recommended to avoid chronic use of the drug in people over 65 years of age, except when other alternatives are not effective or when patients can take gastroprotective drugs (proton pump inhibitors or misoprostol). Drugs from the group of NSAIDs increase the risk of gastrointestinal bleeding or peptic ulcers in patients at an increased risk, including the use of both antiplatelet drugs and anticoagulants. The risk can be reduced with the use of proton pump inhibitors or misoprostol, but cannot be eliminated. They can also increase blood pressure and lead to kidney damage. The risk is related to the dose [13]. During a study conducted in Germany, Nguyen et al. identified a prevalence of NSAID prescription of 45.4% in the examined population of 284 patients aged 65–89 years [35].

Within the 2019 AGS Beers criteria, individual drugs are not listed, which is a limitation, for example, the first- and second-generation antipsychotics are listed, but without listing individual drugs, which makes it difficult to use the criteria. Also, diuretics are listed as drugs that need to be used with caution without highlighting individual diuretics, which is why we interpreted and listed them all as PIMs, even though in most cases their prescription is justified. The next example is represented by benzodiazepines, which according to the 2019 AGS Beers are listed as drugs to be avoided, without recommendations about the length of administration and dose adjustment. According to the above, every patient who was treated with benzodiazepines was identified as a PIM patient, regardless of the length of administration and the dose. Also, from the group of benzodiazepines, the drug bromazepam is not listed in the 2019 AGS Beers list, although the recommendation, explanation and quality of evidence refer to the entire group of benzodiazepines. During this research, we assessed the drug as potentially inadequate. It is probably not listed in the 2019 AGS Beers list because it is not registered on the United States’ market by the Food and Drug Administration.

Our study had several limitations. The research was conducted at three wards, which may not give results that could be generalized to all wards in the UCCRS. However, through the assessment of pharmacotherapy, patients’ chronic therapy was also analyzed, so the results can probably be generalized. After the hospital discharge, the number of PIMs decreased, which we viewed as an accepted intervention, but certainly a study design is needed that will monitor the entire course of treatment and the exact reason for the doctor stopping the drug.

Most of the other studies were performed retrospectively, so not many similar studies involving clinical pharmacists in reducing PIMs have been published. Therefore, we believe that the contribution of this research is significant for the optimization and improvement of pharmacotherapy for the elderly population, especially in low-income settings.

## 5. Conclusions

This pilot study indicates a greater risk of potentially inadequate prescribing in elderly patients where polypharmacy is frequent. Potentially inadequate prescribing results in a higher risk of interactions and adverse drug reactions, and the optimization and individualization of pharmacotherapy is necessary. A large, well-designed study needs to be conducted in order to confirm our findings and establish the role of clinical pharmacists in low-resource settings, where this is needed even more than in HICs.

## Figures and Tables

**Figure 1 pharmacy-12-00084-f001:**
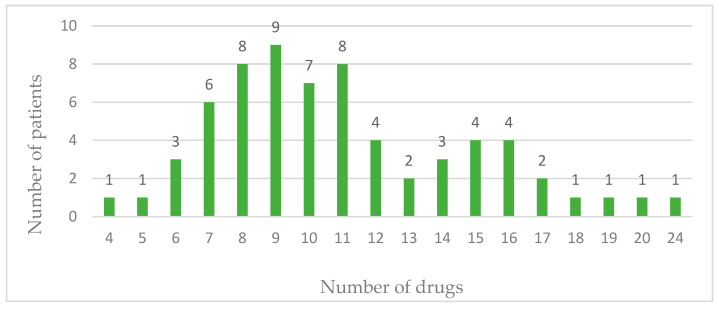
Number of drugs.

**Figure 2 pharmacy-12-00084-f002:**
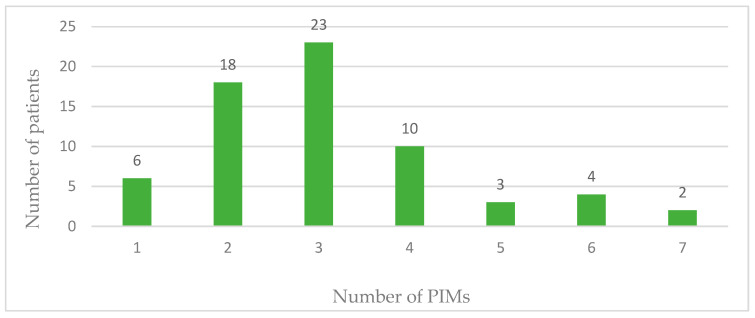
Number of potentially inappropriate drugs per patient.

**Table 1 pharmacy-12-00084-t001:** American Geriatrics Society principles for optimal use of the 2019 AGS Beers criteria [15].

1	Medicines according to the Beers criteria are potentially inappropriate, not definitely inappropriate.
2	Read the explanation of the recommendation for each criterion. The warnings and instructions given there are important.
3	Understand why drugs are included in the list according to the Beers criteria and adjust the approach to drugs accordingly.
4	Optimal application of the Beers criteria includes the identification of potentially inappropriate drugs and, where possible, offers safer non-pharmacological and pharmacological therapies.
5	The Beers criteria should be the starting point for a comprehensive process of identifying and improving the suitability and safety of medicines.
6	Access to drugs included in the Beers criteria should not be unduly restricted by prior authorization and/or health plan environmental policies.
7	Beers criteria are not equally applicable to all countries.

**Table 2 pharmacy-12-00084-t002:** Patients’ demographic data.

Characteristic	Value
Age, years, mean ± SD	76.31 ± 7.04
Sex, male, n (%)	36 (54.50)
I age group (65–75 years), n (%)	33 (50.00)
II age group (>75 years), n (%)	33 (50.00)
Neurology, n (%)	31 (46.97)
Cardiology, n (%)	16 (24.24)
Pulmology, n (%)	19 (28.79)

**Table 3 pharmacy-12-00084-t003:** Distribution of prescribed medicines.

Characteristic	Value			
	Neurology (N = 31)	Cardiology (N = 16)	Pulmology (N = 19)	Total
Number of prescribed medicines, n (%)	307 (42.23)	179 (24.62)	241 (33.15)	727 (100.00)
Female, n (%)	166 (22.83)	79 (10.87)	83 (11.42)	328 (45.12)
Male, n (%)	141 (19.39)	100 (13.76)	158 (21.73)	399 (54.88)
I age group (65–75 years), n (%)	152 (20.91)	99 (13.62)	124 (17.06)	375 (51.58)
II age group (>75 years), n (%)	155 (21.32)	80 (11.00)	117 (16.10)	352 (48.42)

**Table 4 pharmacy-12-00084-t004:** Distribution of PIMs.

	Neurology	Cardiology	Pulmonology	Total PIMs
Female, n (%)	46 (22.55)	22 (10.78)	20 (9.80)	88 (43.14)
Male, n (%)	41 (20.10)	29 (14.22)	46 (22.55)	116 (56.86)
I age group (65–75 years), n (%)	39 (19.12)	26 (12.75)	32 (15.67)	97 (47.55)
II age group (>75 years), n (%)	48 (23.53)	25 (12.25)	34 (16.67)	107 (52.45)
PIMs, n (%)	87 (42.65)	51 (25.00)	66 (32.35)	204 (100.00)

PIMs—potentially inappropriate medicines.

**Table 5 pharmacy-12-00084-t005:** Correlation of number of prescribed drugs with number of PIMs.

	Mean (±SD)	r	*p*
Number of drugs	11.02 (3.94)	0.59	<0.01
Number of PIMs	3.09 (1.42)		

r—Pearson correlation coefficient.

**Table 6 pharmacy-12-00084-t006:** Distribution of PIMs according to ATC classification system.

ATC Class of Medicines		Number of Medicines (%)
C	Medicines that affect the cardiovascular system	91 (44.61)
A	Medicines that affect the alimentary tract and metabolism	41 (20.10)
N	Medicines that affect the nervous system	31 (15.19)
M	Medicines that affect the musculoskeletal system	28 (13.73)
B	Medicines with effects on blood and blood organs	7 (3.43)
H	Hormonal preparations for systemic use, excluding sex hormones and insulin	5 (2.45)
J	Antiinfectives for systematic use	1 (0.49)

ATC—anatomical–therapeutic–chemical.

**Table 7 pharmacy-12-00084-t007:** List of PIMs.

Drug	Number of PIMs, n (%)
Furosemide	37 (18.14)
Pantoprazole	34 (16.67)
Ketoprofen	23 (11.27)
Spironolactone	22 (10.78)
Bromazepam	20 (9.80)
Hydrochlorothiazide	10 (4.90)
Digoxin	9 (4.41)
Diazepam	5 (2.45)
Metoclopramide	5 (2.45)
Amiodaron	4 (1.96)
Diclofenac	4 (1.96)
Enoxaparin	4 (1.96)
Indapamide	3 (1.47)
Methylprednisolone	3 (1.47)
Acetylsalicylic acid	2 (0.98)
Amiloride	2 (0.98)
Glimepiride	2 (0.98)
Methyldopa	2 (0.98)
Sertraline	2 (0.98)
Ciprofloxacin	1 (0.49)
Dexamethasone	1 (0.49)
Enalapril	1 (0.49)
Haloperidol	1 (0.49)
Clonazepam	1 (0.49)
Nimesulide	1 (0.49)
Paroxetine	1 (0.49)
Prednisone	1 (0.49)
Promazine	1 (0.49)
Ramipril	1 (0.49)
Rivaroxaban	1 (0.49)

PIMs—potentially inappropriate medicines.

**Table 8 pharmacy-12-00084-t008:** Medians of the number of PIM.

Potentially Inappropriate Medicines	Median (IQR)	*p* Value
Before intervention	3 (2–4)	<0.01 *
After intervention	2 (1–3)	

IQR—interquartile range; * Wilcoxon test.

## Data Availability

Data supporting reported results can be obtained from the corresponding author.

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
