# Peer review of "The Role of a Clinical Pharmacist in the Identification of Potentially Inadequate Drugs Prescribed to the Geriatric Population in Low-Resource Settings Using the Beers Criteria: A Pilot Study"

_pharmacy, 2024, doi:10.3390/pharmacy12030084_

Round 1

Reviewer 1 Report

Comments and Suggestions for Authors

In this study to identify PIPs in the therapies of hospitalized older patients, the results are clearly reported and they confirm the findings of other similar studies conducted in older patient populations. These findings support the importance of having a clinical pharmacist on the multidiscplinary care team, particularly for older patients with comordibidity and polypharmacy.

I only have a few suggestions for the authors:

- please specify the version of the Beers criteria used, both in the abstract and method section. Authors should also justify why they didn't use the 2023 version of the Beers criteria;

- authors should standardize the number of significant digits (percentages) in the introduction section;

- I don't understand the meaning of the sentence starting with "Potentially inappropriate medication (PIM) is medication with which use the risk of..." in the intruduction section. Please correct the sentence;

- in the result section some number are written in letters (after Figure 1), please correct;

- Finally, I have a curiosity: how were inappropriate drugs considered according to Beers criteria in the elderly meeting certain conditions, such as blood electrolyte levels? Did the authors have all the necessary information?

Comments on the Quality of English Language

Minor English revision is required, particularly some articles are missing.

Reviewer 2 Report

Comments and Suggestions for Authors

Dear Authors,

Please specify the scientific sources that were used to compile the Study Design and Settings (2.1) and Patients and Methods (2.3) sections. Additionally, kindly include the study period in section 2.1. Clarify the ethics permit information, including the issuing authority and date.  Provide a detailed description of the results, including Tables 4, 5, and 6.

Reviewer 3 Report

Comments and Suggestions for Authors

      1.            As the authors mention, the article aims to provide information on the impact of the interventions of the clinical pharmacist in a health system where the clinical pharmacy is at the beginning. Unfortunately, this aspect is the least analyzed in the article. In my opinion it is necessary to explicitly present the interventions of the clinical pharmacist that led to the decrease of PIM. Also, the interventions accepted and those rejected by doctors, as well as the reasons for the rejection.

      2.            I think it is necessary to specify the reason why the authors chose to apply the Beers criteria, although that list initially was only intended for use in the United States, given that there are other tools for identifying PIM used in Europe? An example is the EU(7)-PIM list, a list of potentially inappropriate medications for older people consented by experts from European countries.

      3.            Some countries have adapted the Beers criteria to their own context, and other countries have observed that the listed drugs may not be applicable in their country. In this context, I think it is necessary to mention the barriers that the authors encountered in using this tool. It is also necessary to mention all the limitations of the study.

      4.            It is necessary to specify when the study was carried out, considering that the authors used the 2019 Beers criteria, but the 2023 update is now available.

      5.            Considering that it is a pilot study, carried out on a small number of subjects, the statistical analysis has little relevance. Thus, it would have been useful to discuss the research results in more detail.

      6.            I recommend revising the references list because some sources, especially the Internet ones, are incorrectly cited.

Reviewer 4 Report

Comments and Suggestions for Authors

This paper aimed to study the role of a clinical pharmacist in the identification of potentially inadequate drugs prescribed to geriatric population in the low resource setting using the Beers criteria. This is a relevant topic since clinical outcome and toxicity of medicines depends on the correct selection and administration. The drugs interactions, particularly in the elderly population polimedicated, and the adverse events associated has a significant economic and social impact.

I would like to see a few questions clarified:

How the sample size was decided, to obtain relevant results?

If possible, improve the presentation of Figure 1, on the x-axis there should be only number of drugs (remove per patients).

If possible, present the results of PIMs vs Number of drugs per patient.

Add limitations of the study and include how the authors predict the extrapolation of data to other countries.

Reviewer 5 Report

Comments and Suggestions for Authors

Dear Authors,

The authors present an interesting and very useful study in the medical community by highlighting the increasing importance of the clinical pharmacist in making extremely important decisions in therapeutics in general, even more so for vulnerable groups of patients, such as the geriatric segment. It is remarkable the concern for the expansion of the complex medical team in making decisions to improve the regimes and therapeutic schemes of the elderly in the society of Bosnia and Herzegovina.

In the format presented for review, it was easier for reviewers to follow if line numbering was applied on the right margin of the article.

In section 2.3. Patients and methods, I suggest the authors to add to the general inclusion criteria and the hospitalization period of the patients taken into the study, especially considering that it is specified that the adoption of the clinical pharmacist's suggestion was observed at discharge, where the number of potentially inappropriate drugs before and after was compared intervention, i.e. before and after clinical pharmacist involvement in therapy assessment. So that the question arises after what period did the adverse reactions appear that led to the conclusion of PIM?

According to table no. 3, the authors state that the number of drugs administered to each patient varied between 4 and 24. For greater clarity, I suggest the authors specify the period in which these drugs were administered, 24 drugs were administered during the entire period of hospitalizations?

Compared to other studies presented in the article (Zhang, Perekh, etc.), the number of patients studied by the authors is extremely small, also the evaluation period is not included, so the question can be asked how representative these results are in order to draw pertinent, general conclusions.

In my opinion, this type of analysis and studies are very useful and should be encouraged, specifying that the number of patients should be expanded to present much more conclusive results and to limit the subjectivity factor.

Also, the authors should include in tabular format the drugs administered to the patients studied. The somehow too general presentation of the evaluated medication requires these clarifications for a pertinent conclusion.

I suggest the authors to include these additional data to support the statistical data presented, otherwise the study is interesting, but incomplete.

The Beers criteria are very useful in highlighting the potentially inappropriate use of medications in older adults. These criteria also include lists of recommendations regarding drugs that can be safely prescribed to adults over 65 years of age. That is why I emphasize the importance of presenting a list of evaluated drugs, or at least by drug class, if this number is very large.

Round 2

Reviewer 2 Report

Comments and Suggestions for Authors

 The manuscript is correct

Author Response

Thank you.

Reviewer 3 Report

Comments and Suggestions for Authors

Thank you for the answers to my comments. 

For more clarity of the analysis, I suggest the authors to present in the Results section a complete list of PIMs identified, not just their distribution according to ATC classification system.

Author Response

Thank you for your kind suggestion. Table 7 with complete list of PIMs has been added to manuscript. Please find it at page 7, line 242.

Reviewer 5 Report

Comments and Suggestions for Authors

Dear Authors,

After these changes, it is an interesting and useful pilot study

Author Response

Thank you.